# When Warm Breaks Cold: Understanding Deacclimations and Reacclimations Cycles as a Key to Winter Crop Resilience

**DOI:** 10.3390/ijms262211080

**Published:** 2025-11-16

**Authors:** Julia Stachurska, Iwona Sadura-Berg, Magdalena Rys

**Affiliations:** The Franciszek Górski Institute of Plant Physiology, Polish Academy of Sciences, Niezapominajek 21, 30-239 Kraków, Poland; j.stachurska@ifr-pan.edu.pl (J.S.); i.sadura@ifr-pan.edu.pl (I.S.-B.)

**Keywords:** *Brassica napus*, climate change, cold acclimation, deacclimation, phytohormones, plant metabolism, reacclimation

## Abstract

Plants such as winter crops are able to acclimate to low temperatures through complex physiological and biochemical modifications that enhance their frost tolerance. Cold acclimation involves changes in, e.g., photosynthetic efficiency, carbohydrate metabolism, the accumulation of osmoprotectants, the remodelling of membrane lipid composition, and the activation of the antioxidant system. Now, due to ongoing global climate change, temperature fluctuations have become more frequent, particularly during the autumn–winter period. Episodes of warm breaks (mainly above 9 °C) during winter disrupt the cold acclimation process and induce deacclimation, leading to a decrease in frost tolerance and a partial reversal of cold-induced metabolic adjustments. However, deacclimation is not just the reversal of acclimation, as evidenced by distinct responses in metabolites and hormones. Moreover, plants are able to regain lost freezing tolerance through reacclimation upon re-exposure to low temperatures. The article aimed to summarize the current knowledge on the basics underlying cold acclimation, deacclimation, and reacclimation. An explanation of these processes is crucial for protecting winter crop plants under the increasing frequency of variable temperatures during their growth.

## 1. Cold Acclimation of Crop Plants

Low temperatures represent a major abiotic stress affecting plant growth, development, and survival, especially in temperate and tropical regions. Cold stress is typically classified into freezing stress (below 0 °C) and chilling stress (0–15 °C). Freezing is common in temperate zones, while chilling stress occurs more frequently in tropical and subtropical areas [1,2].

Cold stress affects both vegetative and reproductive stages and leads to chlorosis, wilting, growth retardation, and, in severe cases, plant death. At the cellular level, cold stress alters membrane fluidity, lipid composition, ion balance, cytoskeletal structure, protein stability, and photosynthetic efficiency [3,4].

Cold acclimation (CA) is a physiological process in which plants increase their freezing tolerance following long-term exposure (for example, three to six weeks) to low, non-freezing temperatures (typically 2–5 °C). This response, observed in winter and also spring crops, involves coordinated metabolic, molecular, and physiological changes that enhance both chilling and freezing tolerance [1,5,6,7]. The metabolic changes associated with CA are quite well known, and they are briefly described below.

### 1.1. Changes in Photosynthesis During Cold Acclimation

A rapid decline in temperature requires immediate adjustments in photosynthetic light reactions and central carbohydrate metabolism to avoid metabolic imbalances that can trigger the generation of reactive oxygen species (ROS), ultimately leading to cellular damage or death [8,9].

Cold temperatures significantly inhibit photosynthesis and alter the expression of photosynthesis-related genes, affecting overall photosynthetic efficiency. Cold exposure can also result in structural modifications, including a reduction in the size of the PSII antenna complex [8]. Within minutes of temperature decline, plants employ photoprotective mechanisms—including state transitions and enhanced non-photochemical quenching—to alleviate PSII excitation pressure. These responses are accompanied by the transcriptional downregulation of photosynthetic proteins and are further refined by the production of isoforms with temperature-optimized functions and modifications in protein activation states. The regulation of the photosynthetic apparatus through redox signalling networks is considered essential for acclimation to chilling stress. CA allows plants to adjust their photosynthetic temperature optimum, enabling them to sustain rates similar to those at normal temperatures [8,10]. Moreover, according to [11], plants acclimated to low-temperature and high-light conditions have a higher electron transport, Calvin–Benson Cycle, and ATP synthesis than non-acclimated plants.

Rapacz et al. [12] have studied different genotypes of barley, and they have found that, in genotypes with higher high-light tolerance, CA significantly increased photosynthetic capacity during high-light exposure. In the case of studies on winter rye and wheat seedlings [13], CA increased the ratio of light-saturated photosynthesis to dark respiration by 2–3 fold in both winter rye and wheat seedlings. This boost was more pronounced in rye.

### 1.2. Carbohydrate Metabolism During Cold Acclimation

Soluble sugars, which are key products of photosynthesis, play a central role in modulating plant growth and development. The initial carbon products exported from chloroplasts are triose phosphates—specifically, glyceraldehyde-3-phosphate and dihydroxyacetone phosphate. These compounds are subsequently converted into hexose phosphates by aldolase. Hexose phosphates then serve as precursors for sucrose biosynthesis, with sucrose being the primary form of sugar transported from photosynthetically active leaves [11].

Soluble sugars, including sucrose, glucose, fructose, and raffinose family oligosaccharides (RFOs), play key roles in cold stress responses. They serve as osmoprotectants, cryoprotectants, energy sources, and signalling molecules [4,14].

During CA, genes related to sugar biosynthesis and transport are upregulated. In *Arabidopsis thaliana*, this includes sucrose phosphate synthetase (SPS) and δ-1-pyroline 5-carboxylase synthetase (P5CS) [15]. In winter rye (*Secale cereale* L.), CA causes significant increases in glucose, fructose, nystose, and 1-kestose, especially after 3 weeks of exposure [16]. Similarly, Padhiar et al. [17] reported an enhanced sucrose accumulation in cold-acclimated chickpea (*Cicer arietinum* L.) compared to non-acclimated controls. During freezing stress, sugars accumulate in both the cytoplasm and the apoplast, contributing to osmotic regulation as cells experience dehydration due to extracellular ice formation. As ice develops in the apoplast, water is drawn out of the cytoplasm, leading to cellular desiccation. In response, sugars like sucrose and raffinose mimic water molecules by forming hydrogen bonds with membrane phospholipids and proteins, thereby stabilizing membrane structures and preserving protein function. This mechanism also helps protect the photosynthetic machinery under freeze-induced dehydration stress [18]. Raffinose protects photosynthetic proteins and prevents protein aggregation, which is critical for overwintering crops like winter wheat and oilseed rape [19]. Sugars also mitigate cold-induced oxidative stress. They directly scavenge ROS and enhance the activity of antioxidant enzymes such as superoxide dismutase (SOD), catalase (CAT), and ascorbic acid peroxidase (APX) [20,21,22]. Furthermore, sugars function as signalling molecules, regulating the expression of cold-responsive genes via transcription factors like *CBF*s [23].

### 1.3. Role of Non-Sugar Osmoprotectants in Cold Stress Tolerance

In addition to sugars, plants accumulate other osmoprotectants such as proline, glycine betaine (GB), polyols, and polyamines during cold stress.

Proline stabilizes membranes and enzymes, scavenges ROS, and interacts with cold signalling pathways, including abscisic acid (ABA) and jasmonic acid (JA) [24,25,26]. Treating cold-sensitive plants with exogenous proline has been shown to improve survival under low-temperature stress, as evidenced by reduced electrolyte leakage and the elevated activity of antioxidant enzymes [27,28].

GB, a naturally occurring compound, is essential for plant adaptation to various abiotic stresses, including cold and freezing conditions. It can accumulate to high concentrations in the cytoplasm without disrupting normal metabolic processes. During cold-induced osmotic stress, GB contributes to maintaining turgor pressure by equilibrating the water potential between the cytoplasm and the apoplast [29]. It is also known that GB maintains membrane integrity and enhances antioxidative defence under freezing conditions [30,31]. Moreover, the external application or genetic enhancement of GB improves cold tolerance [32,33].

Polyols (sugar alcohols, like mannitol and sorbitol) are derived from sugars such as fructose and glucose, and they can perform multiple protective functions in plants. They help regulate osmotic balance, stabilize macromolecular structures, and reduce the damage caused by oxidative stress [34]. Polyols like mannitol regulate osmotic balance and act as ROS scavengers [35]. An exposure to cold results in elevated sorbitol levels in leaf and phloem tissues, aiding in osmotic regulation and contributing to the prevention of ice formation [36].

Polyamines such as putrescine, spermidine, and spermine contribute to cold stress tolerance by stabilizing membranes, modulating hormone pathways (e.g., ABA), and regulating stress gene expression [37,38].

### 1.4. Water Management During Cold Acclimation

A well-documented response of plants to cold exposure is water loss, which plays a significant role in the development of frost tolerance. Generally, plants with reduced water content exhibit a greater resistance to freezing stress, as intracellular water loss minimizes the risk of ice crystal formation within cells [39]. Key indicators used to evaluate plant–water interactions include the relative water content (RWC), leaf water potential, transpiration rate, and stomatal conductance, as well as water use efficiency (WUE) [40]. A useful parameter is also water band index (WBI). This is a spectral index calculated from leaf reflectance measurements. It quantifies the relative water content in plant leaves, with lower WBI values indicative of a reduced leaf water status [41]. In cold-acclimated plants, RWC typically declines as part of an adaptive response aimed at reducing cellular water content and enhancing freezing tolerance. For instance, Yu, X. et al. [42] reported a significant decrease in RWC in two cold-acclimated rice (*Oryza sativa* L.) cultivars compared to non-acclimated controls. Similar results were obtained by [43] in oilseed rape (*Brassica napus* L.) plants following a three-week CA period at 4 °C.

The proteins involved in cellular water transport, such as aquaporins, play a key role in regulating the dehydration of plant tissues [44]. Aquaporins are essential for maintaining water balance in plant cells, as they allow rapid and selective water transport across membranes, especially during dehydration or water stress. They are classified into several subfamilies, among which plasma membrane intrinsic proteins (PIPs) and tonoplast intrinsic proteins (TIPs) are especially relevant for water transport in roots and leaves [45,46]. Rys et al. [43] found that the abundance of the plasma membrane aquaporin BnPIP1 increased in leaves of cold-acclimated oilseed rape cultivars following a three-week exposure to 4 °C. Similarly, Aroca, R. et al. [47] reported elevated levels of PIP1 proteins and phosphorylated PIP2 isoforms in the roots of both chilling-tolerant and chilling-sensitive maize cultivars subjected to 5 °C for three days.

In the case of the accumulation of *PIP* transcripts, it is often observed that chilling causes a significant reduction—ranging from two- to fourfold—in the expression levels of most *PIP* transcripts. This effect was observed, for example, in maize [47], oilseed rape [43], barley [48], *Festuca* [49], *A. thaliana* [50], and rice [51]. This apparent divergence between transcript and protein levels suggests the involvement of post-transcriptional and post-translational regulatory mechanisms in aquaporin-mediated water transport during CA.

### 1.5. Changes in Plant Membranes

Cell membranes are among the first targets of cold stress. Low temperatures are known to alter the lipid composition of cell membranes by increasing the proportion of unsaturated fatty acids, a change associated with reduced membrane rigidification [52,53,54]. This prevents the loss of fluidity and phase transitions that impair membrane function.

Specific phospholipids, including phosphatidylcholine (PC) and phosphatidylethanolamine (PE), can be modified to reduce the phase transition temperature of cellular membranes, thereby inhibiting the formation of gel phases that compromise membrane function [55]. Additionally, adjustments in the sterol-to-phospholipid ratio help regulate membrane permeability and enhance stability under freezing stress [56]. As previously noted, osmoprotectants accumulate in response to cold stress and contribute to membrane stabilization by substituting for water molecules around phospholipids, which prevents membrane fusion and leakage during freeze-induced dehydration [57].

### 1.6. Changes in Protein Content

It is well known that, as a result of CA, several protective proteins such as late embryogenesis abundant (LEA) proteins, cold shock proteins (CSPs), heat shock proteins (HSPs), and antifreeze proteins (AFPs) accumulate in plants, contributing to cellular cryoprotection [58].

LEA proteins, often categorized as hydrophilins due to their high hydrophilicity and shared structural characteristics, are predominantly considered intrinsically disordered proteins [59]. Among them, COR15A is a well-studied example that associates with membrane surfaces during dehydration, playing a key role in stabilizing membranes under freezing stress conditions [60]. Numerous LEA proteins have been identified across various plant species, where they contribute significantly to enhancing tolerance to cold and freezing environments [61,62].

AFPs, by contrast, interact directly with the surfaces of ice crystals to suppress their growth [63]. In plants, AFPs exhibit a strong Ice Recrystallization Inhibition (IRI) activity, suggesting that this function is particularly vital in their adaptation to cold stress [64,65]. Some plant AFPs have also been identified as key components in improving freezing tolerance [66,67].

CSPs, which contain a single cold shock domain (CSD), act as RNA chaperones in both bacteria and plants [68,69]. In *Arabidopsis*, CSP2 and CSP3 have been shown to play critical roles in modulating the plant’s response to freezing temperatures [70,71].

HSPs not only accumulate in plants exposed to high-temperature stress but are also induced in response to other abiotic stresses, including low-temperature conditions [5].

Recent studies have explored the role of HSPs, particularly HSP70 and HSP90, in cold-acclimated crops. Cabane et al. [72] reported enhanced HSP70 levels in soybean cultivars subjected to moderate cold, while Krishna et al. [73] observed an increased HSP90 accumulation in spring oilseed rape exposed to low temperatures. Similar trends were found in rice seedlings, where HSP70 and HSP90 were significantly upregulated under cold stress [74,75,76,77].

In potato, prolonged exposure to 4 °C led to increased HSP70 levels in two cultivars [78]. Winter wheat showed rising HSP70 levels during early CA, while HSP90 decreased at later stages [79]. Barley and oilseed rape studies further highlighted cultivar-specific responses: some varieties showed a strong HSP accumulation under cold, while others exhibited reduced levels compared to controls [80,81,82].

Previous studies have demonstrated that CA changes the accumulation of stress-related proteins, including LEA proteins, AFPs, COR proteins, and HSPs, which are involved in protecting cellular structures under freezing conditions. It can be assumed that the accumulation of stress-related proteins during CA is not only a passive consequence of low-temperature exposure but represents an integral part of a coordinated adaptive response [5,66]. For example, LEA and COR proteins not only act as structural stabilizers but may also modulate intracellular water dynamics and membrane phase transitions, thereby influencing signal transduction efficiency under freezing stress.

Similarly, the induction of HSP during cold exposure likely serves functions beyond their classical chaperone activity [5]. HSP could stabilize components of signal transduction pathways, maintain proteostasis in organelles such as chloroplasts and mitochondria, and facilitate the refolding or degradation of regulatory proteins involved in CA. Their differential expression patterns among species and cultivars suggest that specific HSP isoforms might contribute to distinct phases of cold response.

It can therefore be hypothesized that HSPs, together with LEA proteins, AFPs, and COR proteins, can form an integrated network that links protein protection with stress signalling and transcriptional regulation. This coordination may allow plants to adjust their physiological state during prolonged cold exposure, optimizing both protection and metabolic efficiency. The activity of these protein networks is closely integrated with hormonal regulation, which plays a crucial role in coordinating plant responses during CA.

### 1.7. Changes in Phytohormonal Homeostasis

Plants adapt to low-temperature stress by precisely modulating the levels and signalling pathways of various phytohormones, which act as internal messengers integrating environmental cues with growth, development, and stress responses [53]. Cold stress alters both hormone levels and signalling dynamics, triggering the cascades essential for CA. The main hormonal changes associated with CA are outlined below.

ABA is rapidly induced by cold and promotes the expression of *CBF* and *COR* genes [83]. Upon exposure to low temperatures, endogenous ABA levels increase rapidly in cold-acclimated *A. thaliana*, wheat (*Triticum aestivum* L.), and oilseed rape plants (*B. napus* L.) [37,82,84]. Exogenous ABA increases freezing tolerance, while ABA-deficient mutants show a reduced acclimation [85,86]. Importantly, ABA contributes to the upregulation of *CBF* (C-repeat binding factor) genes involved in osmoprotection, antioxidant defence, and membrane stabilization, which then induce the expression of cold-responsive (*COR*) genes, many of which encode proteins involved in cryoprotection, such as dehydrins and LEA proteins [87,88].

JA plays a vital role in the plant defence mechanism. The endogenous JA level is increased due to cold treatment in wheat [89], rice [90], and oilseed rape [82]. JA signalling is implicated in the early stages of CA. JA enhances cold tolerance by activating *CBF* genes and promoting ROS defence [91,92].

Gibberellins (GAs) are important growth-promoting phytohormones that are known to be suppressed during CA. The cold-induced or constitutive expression of *CBF1* (C-repeat binding factor 1) in plants leads to a reduction in the levels of bioactive GAs, primarily by upregulating the expression of GA-inactivating enzymes such as GA2ox3 and GA2ox6. This decrease in active GAs results in the stabilization and accumulation of DELLA proteins, which are nuclear growth repressors whose degradation is normally promoted by GAs. The accumulation of DELLAs under cold or *CBF1*-expressing conditions contributes to growth restraint, such as dwarfism and delayed flowering, and also enhances freezing tolerance through mechanisms that are partly independent of the classic *CBF* regulon. Furthermore, there is evidence of a feedback loop where DELLAs can also promote the expression of *CBF* genes, involving interactions with jasmonate signalling pathways, thus reinforcing the plant’s adaptation to cold by coordinating growth suppression and stress tolerance responses [93,94,95,96].

Auxin, particularly indole-3-acetic acid (IAA), is a key hormone known for its role in regulating plant growth and development. Beyond these functions, auxin also contributes to the regulation of plant responses under cold stress. An exposure to low temperatures can interfere with auxin biosynthesis, transport, and signalling, leading to inhibited root and shoot growth during chilling conditions [97]. Typically, cold stress results in decreased auxin levels and altered distribution patterns, which serve as an adaptive mechanism to reduce growth and conserve energy. Auxin also interacts with other hormonal pathways, including ABA and ethylene, to coordinate the gene expression involved in cold tolerance [98]. Recent findings indicate that the regulation of auxin transporters, such as PIN proteins, and auxin-responsive genes contributes to CA by affecting cell division and expansion processes [97]. Furthermore, auxin signalling may intersect with cold-response pathways through transcription factors like *ICE1* and *CBF*s, integrating hormonal regulation with cold-inducible gene networks [99].

Salicylic acid (SA) contributes to cold tolerance by increasing antioxidant activity and regulating *CBF* expression. In *Arabidopsis*, SA-deficient lines show impaired CA, highlighting its functional importance [100].

Brassinosteroids (BRs), a group of plant steroid hormones, contribute significantly to both inherent and inducible freezing tolerance in plants. Their role in enhancing cold tolerance is linked to their ability to stabilize the photosynthetic apparatus and regulate the activity of ROS-scavenging enzymes, such as SOD, CAT, and peroxidases (POD), thereby protecting cells from oxidative damage under low-temperature conditions [101]. In addition to these protective effects, BRs influence key cold-responsive regulatory pathways, particularly the *ICE1–CBF–COR* cascade, through transcription factors like *BZR1* and *CES*, which serve as important nodes connecting BR signalling to cold stress responses [102]. Pociecha et al. [16] have found that CA (3 and 6 weeks at 4 °C) caused a significant increase in castasterone content in winter rye plants, in comparison to non-acclimated plants. Sadura et al. [103] studied barley (*Hordeum vulgare* L.) cv. Bowman and its mutants with disturbed BR biosynthesis BW084 and disturbed BRs perception BW312. They have found that, after CA, the BW312 mutant and its wild type Bowman were characterized with a significantly higher content of castasterone and 28-homocastasterone than before CA. Functional studies on *Arabidopsis* have demonstrated that mutants with an impaired BR perception (e.g., *bri1*) are more susceptible to cold, while BR-overproducing lines show an improved cold and freezing tolerance [104]. Furthermore, cultivars of wheat with higher endogenous BR levels have been associated with greater frost resistance [105]. These hormones also contribute to CA by modifying membrane composition and promoting the accumulation of compatible solutes such as proline and soluble sugars, which support osmotic balance and cellular protection during stress [101].

Ethylene plays a dual role. It may promote cold tolerance by enhancing antioxidant defences but can also accelerate senescence under prolonged cold stress. Its effect depends on species, timing, and concentration [14,106,107].

Cytokinins (CKs), a group of phytohormones primarily involved in regulating cell division, shoot formation, and the delay of leaf senescence, also contribute to plant responses under cold stress. An exposure to low temperatures often triggers changes in endogenous CK levels, with effects on cold tolerance varying depending on species, tissue type, and developmental stage. In some cases, cold stress leads to a reduction in CK content, which is associated with growth suppression and increased stress resilience [108]. However, the application of exogenous cytokinins has been shown to enhance cold tolerance by delaying senescence, preserving photosynthetic efficiency, and stabilizing membrane structures [109]. CKs participate in hormonal cross-talk, with pathways such as those of ABA and ethylene influencing the expression of cold-responsive genes and promoting the accumulation of osmoprotectants. Notably, CKs regulate stress-responsive transcription factors, including *CBF*, which play central roles in cold adaptation [110]. Moreover, cytokinin response regulators (*ARR*s), especially type-A ARRs, exhibit altered expression under low temperatures, and their overexpression has been linked to an enhanced cold tolerance through the modulation of stress-related gene expression [111].

### 1.8. Antioxidant System

Cold stress is associated with the overproduction of ROS, leading to oxidative damage in plant cells. Antioxidant enzymes such as SOD, CAT, APX, and guaiacol peroxidase (POD) play a critical role in mitigating this damage. SOD catalyzes the dismutation of superoxide radicals (O_2_^−^) into hydrogen peroxide (H_2_O_2_), which is subsequently detoxified by CAT and POD [112].

Numerous studies have demonstrated that CA enhances the antioxidant defence system in plants. For instance, Xu et al. [113] reported increased activities of SOD, CAT, and APX in cold-acclimated citrus seedlings compared to non-acclimated controls. Additionally, transcript levels of *CuZnSOD1*, *CAT1*, and *APX* were significantly upregulated in the cold-acclimated plants. In the study on near-isogenic wheat lines, Baek and Skinner [114] found that CA at 2 °C for 28 days led to the upregulation of *MnSOD* and *t-APX* transcripts, while the expression level of *CAT* was downregulated and, in the case of the expression of *FeSOD* and *Cu*, *ZnSOD* downregulation was observed. Janmohammadi et al. [115] observed elevated SOD, APX, and CAT activities in winter wheat following a 20-day CA at 4 °C, in comparison to non-acclimated plants. In the case of spring wheat, the opposite effect was observed for the SOD and APX activity of cold-acclimated plants in comparison to non-acclimated plants. Doğru and Çakirlar [116] studied winter oilseed rape that was cold-acclimated at 2 °C for 15 days. They have observed a significant increase in the activity of SOD and APX in cold-acclimated plants of both studied cultivars. Similar results were obtained by [117]; namely, they have observed a higher activity of SOD, CAT, and POD in cold-acclimated (at 10/3 °C for 7 days) oilseed rape plants than in non-acclimated plants. Also, in the case of chickpea (*Cicer arietinum* L.) plants that were subjected to cold stress (0 °C for 24 or 48h), a significant increase in the activity of CAT and APX was observed [118]. Similar results were obtained by Padhiar et al. [17], who reported enhanced activities of SOD, CAT, APX, and glutathione reductase (GR) in all organs of a cold-tolerant cultivar during CA, compared to non-acclimated plants.

All the physiological, biochemical, and structural modifications that take place in plants during the process of cold acclimation are schematically summarized and illustrated in Figure 1 below.

## 2. Deacclimation of Crop Plants

The ability of plants to survive low temperatures during winter largely depends on their capacity to undergo cold acclimation—a physiological and biochemical process that increases freezing tolerance in response to gradually decreasing temperatures and shortened photoperiods. During this process, plants accumulate various protective compounds, including soluble sugars, amino acids, and specific proteins, which stabilize cellular structures and maintain osmotic balance under freezing conditions, as described in the previous part [119,120]. However, in recent years, climate change and global warming have significantly altered seasonal temperature patterns, particularly in temperate regions. Winters have become increasingly characterized by frequent warm breaks, during which temperatures may rise above 9 °C and occasionally reach as high as 20 °C for several days [43,121,122].

Such transient periods of warming lead to deacclimation (also called dehardening)—a process that reverses the cold acclimation state and consequently reduces frost tolerance. Deacclimation may occur during late autumn, mid-winter, or early spring, whenever temperatures temporarily exceed the threshold required for maintaining the acclimated state [123]. The rate and extent of deacclimation are influenced by the intensity and duration of the warm period, as well as by the plant species and genotype [43,121]. This process disrupts the natural adaptation cycle of winter crops and has become one of the crucial but widely neglected aspects of plant winter survival under changing climatic conditions [121,123].

Physiologically, deacclimation is associated with a reduction in frost tolerance due to metabolic and structural changes in plant tissues, which will be discussed in detail in this paper. Additionally, deacclimation often results in the resumption of growth processes, including stem elongation and bud development, which further reduce cold resistance [121]. In extreme cases, prolonged warm periods can trigger the premature reactivation of metabolism and photosynthetic activity [43,122].

Recent studies have revealed that deacclimation is not simply the reverse of cold acclimation but rather an independent physiological pathway involving complex regulatory mechanisms at both the biochemical and molecular levels [43,121]. Analyses have shown that the metabolic profile of deacclimated plants becomes more similar to that of non-acclimated controls, highlighting the depth of physiological changes during the process [43].

Given the increasing frequency of mid-winter warm periods predicted by climate models for the northern hemisphere throughout the 21st century (IPCC, 2021) [124], understanding the mechanisms of deacclimation and their impact on frost tolerance and crop yield has become a pressing issue. Winter crops such as oilseed rape (*B. napus* L.), barley (*H. vulgare* L.), and wheat (*T. aestivum* L.) are particularly susceptible, as deacclimation episodes can lead to significant economic losses due to frost injuries following re-cooling periods [43,121]. Figure 2 shows a possible time when periods of higher temperatures can occur, disrupting the cold acclimation and leading to deacclimation and then reacclimation in plants.

Detailed changes occurring during deacclimation are described in the following paragraphs.

### 2.1. Changes in Photosynthesis During Deacclimation

During cold acclimation, plants commonly downregulate growth and tune photosynthetic processes so that light harvesting, electron transport, and carbon metabolism are matched to the low metabolic demand and to the limits imposed by temperature. Cold-acclimated leaves often exhibit reduced rates of net CO_2_ assimilation, shifts in energy partitioning toward photoprotective pathways, changes in chlorophyll fluorescence parameters, and adjustments in dark respiration that together minimize photo-oxidative damage while supporting the synthesis of protective compounds [9,12,125]. When a sudden warm break interrupts the cold period, causing deacclimation, these photosynthetic traits are rapidly reversed. The studies in crops and model species show that deacclimation causes a reactivation of both the light and dark phase of photosynthesis, often within hours to days after warm temperatures [126].

In our works, we proved that, in winter oilseed rape (*B. napus* L.), deacclimation reversed many of the cold-induced changes in photosystem performance: chlorophyll *a* fluorescence (both prompt and delayed signals) and indicators of PSII and PSI efficiency returned toward non-acclimated values after a one-week warm break. These changes were cultivar-dependent and correlated with the degree of loss of frost tolerance, indicating that photosynthetic reactivation during deacclimation is tightly linked to physiological vulnerability to subsequent freezes [122]. Our earlier work in the same crop likewise reported that cold acclimation reduced the intensity of the light reactions of photosynthesis, and that deacclimation tended to restore the photosynthesis intensity, accompanying the observed decline in freezing tolerance [43].

Generally, deacclimation causes a rapid reactivation of photosynthetic processes in crop plants, which, while beneficial for growth in the returning mild conditions, weakens the biochemical and biophysical protections acquired during cold acclimation. The rate of these photosynthetic changes is therefore a key physiological factor determining whether a plant will maintain frost tolerance or suffer damage when cold temperatures return.

### 2.2. Carbohydrate Metabolism During Deacclimation

Carbohydrate metabolism plays a central role in the acquisition and loss of freezing tolerance in plants. During cold acclimation, an exposure to low temperatures leads to the accumulation of soluble sugars, which stabilize cellular membranes and proteins, reduce ice crystal formation, and contribute to osmotic balance and reactive oxygen species detoxification. The synthesis and accumulation of these sugars are tightly linked to the activation of cold-responsive genes and to metabolic reprogramming that prioritizes the production of stress-protective compounds over growth-related processes [127,128,129].

When plants are exposed to a period of higher temperatures during winter, these processes are disrupted, initiating deacclimation. Deacclimation leads to a reduction in the content of soluble carbohydrates, thereby diminishing the plant’s ability to withstand freezing stress. In *A. thaliana*, it has been shown that concentrations of glucose, fructose, sucrose, and raffinose decrease by more than 50% within 24–48 h after plants are transferred from cold to warm conditions [130]. This decrease coincides with the downregulation of cold-responsive genes and with a reactivation of photosynthesis and primary metabolism. Pagter et al. [131] demonstrated that both transcriptomic and metabolic shifts occur within the first day of deacclimation, indicating a rapid sugar catabolism and redistribution of carbon into growth-related pathways.

Comparable observations have been made in cereal crops. In winter wheat and barley, the deacclimation process is characterized by a substantial decline in sucrose, glucose, and fructose levels, which correlates closely with a reduction in frost tolerance [132]. In oilseed crops such as *B. napus*, warm breaks during winter have been shown to significantly reduce the content of soluble carbohydrates, increase osmotic potential, and lower freezing tolerance [43]. Importantly, not all cultivars respond equally: some maintain higher sugar levels after deacclimation, suggesting that genotypic differences in sugar metabolism may contribute to resistance under fluctuating thermal conditions.

The regulation of sugar metabolism during deacclimation involves several interconnected physiological and molecular processes. Warming induces a shift from catabolic to anabolic carbon utilization, stimulating photosynthetic recovery and starch biosynthesis, while decreasing the synthesis of soluble protective sugars [123,131]. The expression of key cold-induced transcription factors such as *CBF*/*DREB1* and *COR* decreases rapidly, leading to a loss of acquired freezing tolerance [131]. During the transition from cold to warm conditions, with changes in cold-responsive transcription factors, sugar transport and allocation undergo substantial reorganization, which is controlled by the expression and regulation of sugar transporter genes. Sucrose transporters (*SUT*/*SUC*), hexose transporters (*HT*), and members of the *SWEET* family facilitate sugar loading, retrieval, and intracellular exchange, and their expression is highly responsive to temperature and carbohydrate content [133,134]. Environmental warming alters the transcription of these transporters, modifying source–sink relationships and influencing how carbon is partitioned between storage and growth. For instance, the overexpression of *AtSWEET4* in *Arabidopsis* enhances freezing tolerance, underscoring a direct functional link between sugar transport activity and cold stress resilience [135].

Taken together, the available evidence indicates that changes in sugar content and carbon management are among the most crucial biochemical determinants of plant winter survival. The loss of soluble sugars during deacclimation weakens multiple protective systems simultaneously, affecting osmotic regulation, antioxidant capacity, and structural stability. In contrast, genotypes with a slower rate of sugar depletion or more efficient reacclimation of carbohydrate metabolism exhibit an enhanced resilience to temperature fluctuations.

### 2.3. Role of Non-Sugar Osmoprotectants in Deacclimated Plants

Among the numerous metabolites involved in plant adaptation to fluctuating temperatures, non-sugar osmoprotectants, particularly proline, polyamines, and glycine betaine, play a central role in maintaining cellular homeostasis during cold acclimation and deacclimation.

Generally, as described in Section 1.3, these compounds contribute to osmotic adjustment, the protection of macromolecules and membranes, and the scavenging of ROS under low-temperature stress [123,136].

During cold acclimation, plants accumulate large amounts of proline and other compatible solutes that stabilize proteins, sustain cell turgor, and support enzymatic activity under dehydration caused by freezing [130]. However, when a warm period occurs during winter, this metabolic balance is disturbed, leading to deacclimation, a process characterized by the reduction in protective metabolites. For example, in *A. thaliana*, proline levels decrease rapidly within the first 24 h of deacclimation, coinciding with changes in gene expression associated with primary metabolism and stress recovery [130,131]. In winter wheat, deacclimation and reacclimation cycles were shown to strongly influence the abundance of free amino acids, including proline and glutamate, linking these metabolites to reversible frost tolerance [132]. Additionally, the dynamics of polyamines (putrescine, spermidine, spermine) in cold-treated *B. napus* have been shown to parallel proline accumulation, emphasizing their shared roles as non-sugar cryoprotectants in regulating membrane stability and hormonal balance during temperature stress [137].

These findings support the hypothesis that the maintenance of non-sugar osmoprotectants such as proline during deacclimation is crucial for preserving residual frost tolerance, and that their rapid depletion may predispose plants to frost injury upon sudden temperature drops. Thus, genotypes with a slower loss or faster re-synthesis of these compounds during transient warming could be better adapted to increasingly variable winter climates.

### 2.4. Water Management During Deacclimation

Water management is one of the most critical physiological processes determining plant survival under fluctuating winter temperatures. During cold acclimation, plants undergo a complex reprogramming of water relations to prevent cellular dehydration and ice formation. This involves osmotic adjustment, changes in cell wall elasticity, and a controlled reduction in membrane water permeability. Such changes minimize extracellular ice propagation and maintain intracellular hydration, ensuring the structural integrity of tissues exposed to subzero temperatures [138,139].

A key element of water homeostasis regulation during cold acclimation and deacclimation is the modulation of plasma membrane aquaporins (AQPs)—integral membrane proteins that facilitate the transport of water and small solutes across membranes [140]. Cold stress typically downregulates AQP expression and activity to restrict transmembrane water movement, thereby maintaining osmotic stability and reducing the risk of intracellular ice nucleation [141]. The results obtained in our earlier study showed that deacclimation induced the reactivation of aquaporin gene *PIP1* in oilseed rape leaves, which could lead to an increased membrane water permeability and enhanced water transport between tissues [43]. The aquaporin regulation during temperature stress is highly plastic and species-, genotype-, and isoform-specific [142]. For example, cold acclimation in rice is accompanied by the upregulation of specific *PIP* genes (*OsPIP1;3*, *OsPIP2;4*) and enhanced water transport, indicating that some *PIPs* are positively associated with cold acclimation [141]. By contrast, studies report decreases in *TIP* or *PIP* transcripts during cold acclimation: in forage grasses (*Festuca* spp.), *TIP1;1* abundance declined with cold acclimation [49]. In *Brassica rapa*, genome-wide surveys show that *PIP* family members respond differentially to low temperatures and other stresses, with some isoforms being cold-inducible and others repressed [143].

### 2.5. Changes in Plant Membranes

Cell membrane permeability is a sensitive physiological indicator of cold tolerance, and its maintenance is central to both cold acclimation and deacclimation processes. During cold acclimation, plants remodel their membrane lipid composition to preserve fluidity and maintain optimal functionality under low temperatures. This involves an increase in the proportion of unsaturated fatty acids, particularly linoleic (18:2) and linolenic (18:3) acids within the phospholipid matrix, which enhances membrane flexibility and prevents phase transitions [144,145]. However, during deacclimation, this lipid remodelling process is at least partially reversed. This phenomenon is associated with a reduction in lipid unsaturation, which could lead to an increased ion efflux and solute leakage. Recently, our study in *B. napus* demonstrated that a one-week warming period following cold exposure significantly decreased the proportion of unsaturated fatty acids in chloroplast membranes, thereby reducing membrane fluidity and stability [146].

Deacclimation also affects the interaction between membranes and associated proteins. The cold-induced accumulation of specific plasma membrane and tonoplast proteins—such as dehydrins, lipid transfer proteins, and COR (cold-regulated) proteins—declines rapidly under warm conditions [147]. Additionally, in *Brassica napus*, the *COR* genes were already downregulated in cold acclimated plants relative to non-acclimated ones; their expression declined even further during deacclimation [148]. These proteins contribute to cell membrane stabilization and cryoprotection during cold acclimation, and their degradation or reduced synthesis during deacclimation further weakens cell membrane integrity and increases permeability in the face of potential subsequent cold stress.

### 2.6. Changes in Protein Content

During deacclimation, when plants shift from cold-acclimated to warmer conditions, the amounts of many proteins change noticeably. Proteins that were accumulated during cold acclimation, e.g., those protecting cells from freezing, start to decrease. At the same time, proteins involved in growth and metabolism become more accumulated, showing that the plant is switching from a stress-protected state back to active growth.

Late embryogenesis abundant (LEAs) proteins, including the cold-regulated (COR) and dehydrin families, are among the key molecular components of cold acclimation and are highly sensitive to temperature shifts during deacclimation. LEA and dehydrin proteins accumulate during cold exposure, where they stabilize membranes, bind water, and protect cellular macromolecules against dehydration and freeze-induced damage. In winter wheat, members of the WCS120 protein family (a group of dehydrins strongly associated with frost tolerance) were accumulated during cold acclimation, and after a five-day deacclimation at 17 °C their levels were significantly reduced. In addition, the degree of reduction correlated with loss of frost tolerance [149]. Another study found that WCS120 accumulation differed between wheat and barley varieties, with genotypes that deacclimated more slowly retaining enhanced levels of WCS120 and exhibiting a better winter survival [150].

As mentioned earlier (Section 1.6), heat shock proteins (HSPs) play essential roles in protecting plants from temperature stress by acting as molecular chaperones that stabilize and refold denatured proteins. During cold acclimation, several HSP families, particularly HSP70, HSP90, and small HSPs (sHSPs), are upregulated and contribute to maintaining protein homeostasis and cellular integrity under low temperatures. During deacclimation, HSP accumulation typically decreases. In our study on winter oilseed rape, three of four cultivars displayed elevated levels of cytoplasmic HSP70 and HSP90 after three weeks of cold acclimation; after a subsequent one-week deacclimation at 16/9 °C (day/night), the HSP70 and HSP90 content most often decreased compared to the cold-acclimated state, indicating that the protective protein accumulation induced during cold is reversed during warm-up and the loss of freezing tolerance [82].

### 2.7. Changes in Phytohormonal Homeostasis

Deacclimation, the process by which plants lose cold-induced tolerance under rising temperatures, is accompanied by changes in hormonal homeostasis. Our recent studies have shown that this process involves complex, dynamic changes in phytohormones levels and in the expression of hormone biosynthesis and signalling genes [82,151]. The general trend observed during deacclimation is a decrease in hormones associated with stress protection, such as ABA, JA, and SA, and a simultaneous increase in hormones that promote growth and development, including gibberellins (GAs), cytokinins (CKs), and auxin (AUX).

During cold acclimation, ABA plays a key role in establishing and maintaining frost tolerance by inducing the expression of cold-responsive genes and promoting the accumulation of protective compounds such as osmolytes and proteins. In our recent studies, we report that, during the deacclimation of cold-acclimated oilseed rape, there was a decline in the ABA content in leaves [82,151].

In cold-acclimated plants, both JA and SA—two key phytohormones involved in stress signalling—are typically accumulated as part of the defence response to low-temperature stress, enhancing the expression of cold- and antioxidant-related genes [152]. However, during deacclimation, their concentrations generally decline, indicating a suppression of stress signalling pathways. In our studies, we demonstrated a significant reduction in JA content following an exposure to warmer temperatures, correlating with a decreased expression of genes associated with jasmonate biosynthesis and signalling [151].

In cold-acclimated plants, GAs levels are generally reduced, supporting the suppression of growth and the maintenance of cold tolerance [93]. However, upon exposure to warmer temperatures, deacclimation induces a reactivation of GA biosynthesis and signalling, reflected by higher concentrations of active GA forms, which we demonstrated [82,151]. Additionally, deacclimation is accompanied by the upregulation of the GA biosynthesis *RGA* gene [151]. We have marked these changes in Figure 3A,B.

This molecular shift enhances the pool of bioactive GAs, promoting the resumption of growth processes previously inhibited during cold acclimation. Additionally, Figure 3C presents the overall changes between the total content of active forms of gibberellins (GA1, GA3, GA4, GA5, GA6, GA7)—growth hormones—and the content of ABA—a stress hormone.

The increase in gibberellin content and decrease in ABA content during deacclimation is a key hormonal adjustment that supports the reversal of cold-induced growth arrest, promoting tissue expansion, meristem activity, and developmental recovery under favourable thermal conditions.

Auxin dynamics during deacclimation commonly reflect a shift from growth suppression toward growth reactivation. Our investigation has shown that, in winter oilseed rape, the changes in the active form of auxin—indole-3-acetic acid (IAA)—are cultivar-dependent. This illustrates that auxin content during warming can vary with genetic background [82,151].

Cytokinins regulate plant growth, cell division, and developmental transitions, and their changing levels play an important role in frost tolerance during cold acclimation and its loss during deacclimation. Generally, cold exposure reduces the content of active CKs, contributing to growth inhibition. During deacclimation, the total CK content increased in oilseed rape cultivars, indicating the hormonal reactivation necessary for cell division and meristem activity. The expression of *BnARR* genes, key components of CK signalling, also increased during deacclimation [151].

These studies demonstrate that deacclimation involves a complex but coordinated shift in hormone balance characterized by a decline in stress-associated hormones (ABA, JA, SA) and a simultaneous increase in growth-promoting hormones (GAs, CKs, IAA).

Brassinosteroids are essential regulators of plant growth and stress tolerance. Cold acclimation is often accompanied by changes in BR levels that help maintain membrane stability, enhance antioxidant defence, and support the synthesis of protective proteins. However, during deacclimation, the BR content and signalling are changing. In winter oilseed rape, the total BR content increased during deacclimation compared to the cold-acclimated state, suggesting a reactivation of BR-dependent growth and metabolic processes. This hormonal rise was accompanied by an elevated expression of BR receptor *BnBRI1*, indicating an enhanced BR perception and signalling capacity [122,153]. Increased BR activity likely interacts synergistically with gibberellins and cytokinins, which also rise during deacclimation, to drive meristem activation and biomass accumulation. Collectively, these findings indicate that BRs are not only components of cold tolerance mechanisms but are also crucial hormonal signals that facilitate the physiological and metabolic recovery of plants upon warming [154].

### 2.8. Antioxidant System

The antioxidant system plays an important role in maintaining redox homeostasis during temperature changes in wintertime. During cold acclimation, plants typically accumulate ROS as signalling molecules that modulate stress-responsive pathways, while simultaneously enhancing the activity of antioxidant enzymes and metabolites to prevent oxidative injury [155]. These include enzymatic antioxidants such as SOD, CAT, and APX, as well as non-enzymatic compounds like ascorbate, glutathione, and phenolic metabolites [156]. However, when plants are exposed to warm periods interrupting cold acclimation, these protective systems are frequently downregulated by a rapid reversal of cold-induced antioxidants, although these changes depend strongly on species, genotype, and the temperatures used. Janmohammadi et al. [115] showed that cold acclimation markedly increased the activities of enzymatic antioxidants like SOD, CAT, APX, and GPX in winter wheat, whereas a subsequent deacclimation caused a pronounced decline in these protective systems and an increase in H_2_O_2_ content. These results are consistent with the more detailed metabolite profiling performed by Vaitkevičiūtė et al. [157] in six winter wheat genotypes. After deacclimation, ascorbate and glutathione pools declined in both leaf and crown tissues, while their redox balance shifted toward more oxidized forms, suggesting a weakening of antioxidant protection and redox homeostasis.

During cold acclimation, plants undergo extensive transcriptional reprogramming of antioxidant-related genes, particularly those of the plastid antioxidant system (PAS), including Cu/Zn-superoxide dismutase (*CSD2*), ascorbate peroxidases (*APX*), or glutathione peroxidase (*GPX*). These genes show differential expression patterns depending on genotypes and environmental conditions. During deacclimation, changes in antioxidant gene expression are usually connected with the reversal of cold-induced patterns. In winter barley, deacclimation-susceptible genotypes show stronger and earlier activation of oxidoreductase and peroxidase genes compared with tolerant genotypes, which exhibit fewer transcriptional changes [158]. This indicates that a rapid and strong antioxidant response during warming may increase the loss of freezing tolerance, whereas a more moderate response helps maintain cold acclimation under transient warm periods.

In general, the deacclimation process is characterized by a rapid decline in antioxidant capacity through a reduction in protective metabolite content, representing a transient loss of ROS homeostasis. These changes are often only partially reversible upon subsequent reacclimation, highlighting the crucial role of antioxidant stability in determining winter survival and the post-stress recovery of crop plants.

All physiological, biochemical, and structural modifications that take place in plants during the deacclimation process compared to the changes occurring during cold acclimation are schematically summarized and illustrated in Figure 4 below.

## 3. Reacclimation of Crop Plants

Deacclimation is a reversible process, and deacclimated plants can regain an increased frost tolerance in a process called reacclimation (RA; rehardening), but its basics remain largely unknown. Reacclimation that follows deacclimation is, generally, a return to the pre-deacclimation level of frost tolerance, but it is not always possible, and the ability to reacclimate decreases with the length of the deacclimation period [159]. Deacclimation can be completely reversed as long as elongation growth has not been initiated. Moreover, deacclimation becomes partly or entirely irreversible when it is associated with the beginning of elongation growth [160]. Interestingly, reacclimation is not the same process as cold acclimation, due to altered plant physiological conditions [161].

The conditions required to regain frost tolerance and metabolic adjustment during the reacclimation process depend on the plant species. Additionally, the photoperiod affects the effectiveness of this process—for instance, wheat cultivated under a 20 h photoperiod exhibited a reduced capacity for reacclimation compared to those grown under an 8 h photoperiod [162]. Generally, winter wheat plants that had lost cold acclimation were able to regain it in a short time after re-exposure to cold conditions [163]. Winter and spring cultivars of oilseed rape can recover full frost tolerance, whereas, in winter wheat, this occurs in only 39% of plants when exposed to progressive cold over 40 days [164]. In contrast, in wheat, 14 days of reacclimation (2 °C) resulted in the regaining of frost tolerance [132]. Another study conducted on winter oilseed rape showed that, after four weeks of reacclimation, plants regained tolerance to a level equal to or greater than that achieved before deacclimation at different temperatures (2/12, 12/2, 12/12, and 12/20 °C), regardless of the photoperiod. In plants deacclimated at 20/12 and 20/20 °C, the ability to reacclimate was diminished by approximately 2 °C [160]. However, another study on oilseed rape shoots showed that a shorter period of reacclimation (4 °C, 5 days) resulted in a restoration of sugar and proline levels similar to cold-acclimated plants. Shoots were able to regain cold tolerance similar to before deacclimation when the deacclimation period did not exceed five days [165].

Similarly to cold acclimation and deacclimation, reacclimation causes physiological and biochemical changes. So far, there are not many research articles about the basics of reacclimation. Those that are already published include alterations in the abovementioned frost tolerance and, among others, sugar and water management, antioxidant activity, and protein accumulation. Also, gene expression patterns differ under temperature decrease during RA. Only 24 h (2 °C) of RA for winter wheat was enough to increase the expression of, e.g., genes related to water deprivation, cold acclimation, and the sterol biosynthetic pathway [166]. Moreover, WRKY-type transcription factors function as positive regulators of frost tolerance [167], and their expression also differs after CA, DA, and RA [166]. Therefore, the findings of [166] indicate that the increase in frost tolerance during CA, DA, and RA is largely controlled by genetic mechanisms upstream of WRKY genes.

Reacclimation is a process affected by cold stress memory and vernalization fulfilment [168].

### 3.1. Changes in Photosynthesis During Reacclimation

Temperature fluctuations during cold acclimation, deacclimation, and reacclimation affect the process of photosynthesis, the expression of genes, and the activity of enzymes connected to this process. Despite the importance of photosynthesis to plants, little is still known about the impact of reacclimation on this process. Studies on oilseed rape revealed that, during reacclimation, the activity of sucrose–phosphate synthase (SPS) increased, since the upregulation of this enzyme is necessary for photosynthesis effectiveness at cold conditions [126,159]. The photosynthetic apparatus of plants of the winter cultivar of oilseed rape was able to reacclimate after DA at a similar efficiency, which was determined by Fv/Fm values [169]. Additionally, a reacclimation of the photosynthetic apparatus is not possible in plants characterized by a higher elongation growth rate, despite the high activity of SPS [126]. A detailed analysis of gene expression patterns revealed that, in reacclimated wheat (freezing-tolerant cultivar Lakaja DS), the genes related to photosynthesis were suppressed, which was in contrast to the enhanced photosynthetic activity during CA [166]. Moreover, in reacclimated wheat, the concentration of chlorophyll a was returned to a level similar to that in cold-acclimated plants [132].

### 3.2. Carbohydrate Metabolism During Reacclimation

A significant part of the metabolic changes occurring during cold acclimation, deacclimation, and reacclimation concerns carbohydrate metabolism. Generally, the concentration of water-soluble carbohydrates in wheat leaves (winter and spring cultivars) increased as a result of reacclimation when compared with deacclimated plants [115]. An increased accumulation of glucose was observed in leaves of reacclimated winter wheat, while, in crowns, a higher level of sucrose was observed. Simultaneously, a decreased concentration of starch was observed in both leaves and crowns [132]. In cold-acclimated oilseed rape leaves, the levels of glucose, fructose, and sucrose increased at least 5-fold, while, after deacclimation, the accumulation of those sugars decreased, but reacclimation increased the abovementioned sugar levels again [164]. In the case of oilseed rape shoots, reacclimation (1, 3, and 5 days at 4 °C) led to a gradual rise in soluble sugar content [165].

### 3.3. Role of Non-Sugar Osmoprotectants in Reacclimated Plants

Proline is an amino acid that is accumulated in plants under stress conditions. Proline acts as an osmolyte, and it is also a part of the antioxidative system [170]. For instance, the proline content in oilseed rape shoots increased as a result of reacclimation [165]. Similarly, in another study, reacclimation increased the concentration of proline in oilseed rape leaves. Interestingly, the content of proline increased rapidly to a level greater than in cold-acclimated plants [171]. The reacclimation of winter wheat also resulted in an increased content of proline, but only in plants that completed the cold acclimation process at constant low temperature [132].

### 3.4. Water Management During Reacclimation

Biochemical and physiological changes underlying cold acclimation, deacclimation, and reacclimation include, among others, sugar and water management. Although the variations in osmotic potential observed after reacclimation did not correspond to differences in the frost resistance of oilseed rape, plants that were less capable of reacclimating typically showed a reduced ability to lower osmotic potential during reacclimation [160]. So far, little is known about the detailed changes in water management during reacclimation. Neither is there any information about changes in the membrane properties of reacclimated plants.

### 3.5. Changes in Protein Content

As was mentioned above, changes in protein composition accompany cold acclimation, deacclimation, and also reacclimation. These changes reflect the plant’s ability to restore the molecular components essential for frost tolerance. For example, changes affect the accumulation of dehydrins, which protect other proteins and membranes against unfavourable structural changes caused by the dehydration that happens as a result of cold acclimation [172]. In winter wheat and both winter and spring cultivars of oilseed rape, RA resulted in the accumulation of dehydrins (47 kDa) reaching a similar level to that observed in cold-acclimated plants [164]. In addition, research on winter wheat demonstrated an increased expression of dehydrin-related genes as a result of RA [166].

Another protein that is inducible by cold and whose accumulation changes upon deacclimation and reacclimation is WCS120—known as a major cold-inducible dehydrin in wheat [149,173]. After the reacclimation of winter wheat, the accumulation of WCS120 was restored only in plants before vernalisation fulfilment. In another study on winter wheat, WCS120 increased significantly in cold-acclimated plants and then decreased in deacclimated plants, and, in reacclimated plants, its level increased to similar to that in cold-acclimated plants [164].

A different pattern of changes was observed in the case of the COR78 protein. COR proteins are positively correlated with maintaining frost tolerance [149]. Its accumulation increased in cold-acclimated oilseed rape plants and decreased after deacclimation and upon reacclimation, and it remained at a non-detectable level [164].

Overall, these findings indicate that, while some cold protective proteins, such as dehydrins, can be reaccumulated during reacclimation, others, like COR78, are not accumulated again [164]. Such selective protein reaccumulation may possibly affect the extent to which plants regain frost tolerance after warm spells during winter.

### 3.6. Changes in Phytohormonal Homeostasis

Plant hormones are key regulators of plant responses to different kinds of stress. An exposure to cold can trigger their biosynthetic pathways, leading to the activation of stress-related signalling mechanisms. Despite the importance of hormonal signalling for plant metabolism, changes during reacclimation are still poorly known.

ABA shows a strong upregulation in response to cold stress. The reacclimation of winter oilseed rape resulted in an increase in ABA content in plants that were previously deacclimated at 2/12, 12/2 and 12/20, 20/2, and these were only marginally higher than after CA. In contrast, spring cultivar showed higher ABA levels after reacclimation than winter plants, with values in most cases approaching those measured before cold acclimation [174]. In another crop plant, winter wheat, reacclimation led to the upregulation of genes involved in ABA-mediated responses [166].

Alterations in the expression of hormone-related genes studied in oilseed rape, such as *BnABF2* (ABA-responsive element-binding factor 2) during deacclimation, are possibly a kind of protective mechanism that helps to regain frost tolerance in reacclimation [151].

Reacclimation also induces alterations in growth-promoting hormones. For instance, in winter oilseed rape, GAs generally reached a similar level as in non-deacclimated plants, except for 12/12 °C-deacclimated plants, which showed a slightly higher content of GAs. In the spring cultivar, reacclimation caused an increase in GA levels across all treatments, including the control [174]. In another experiment of [174], with different times of deacclimation and reacclimation, reacclimation caused an increase in GA content in plants deacclimated at 12/12 °C and 12/20 °C, while, at 20/12 °C and 20/20 °C, a decrease was observed.

### 3.7. Antioxidant System

Antioxidants in plants play a crucial role in protecting cells from the damage caused by reactive oxygen species, thus helping them cope with environmental stresses such as low temperatures. Research on winter wheat has shown that antioxidant levels typically rise during cold acclimation, decrease during deacclimation, and return to levels similar to those in cold-acclimated plants after reacclimation. For example, concentrations of total and reduced glutathione increased in the leaves of cold-acclimated plants, decreased after deacclimation, and increased after reacclimation (14 days, 2 °C) to a similar level as in cold-acclimated plants [157]. Moreover, reacclimation induced some changes in the activity of antioxidative enzymes. In winter cultivars of wheat, reacclimation (10 days, 4 °C) did not affect the activity of SOD when compared with deacclimated plants, but increased the activity of APX and CAT in a more cold-hardy cultivar [115].

A summary of changes induced by reacclimation in comparison with deacclimation-induced changes is presented in Figure 5.

### 3.8. Cold Stress Memory

Despite the lack of a central nervous system, plants can detect, remember, and predict environmental changes. The mechanisms of cold stress memory are still poorly known, but previous studies reveal that they are largely based on genetic basics. Plants show various forms of stress memory, including somatic, intergenerational, and transgenerational types that are controlled by epigenetic mechanisms like DNA and histone modifications, as well as microRNAs [175]. They differ in duration—for example, somatic stress memory lasts for a few days or weeks, while transgenerational memory remains for the next generation, but the inheritance mechanisms are not clear [175]. Some stress-induced modifications return to the non-stress levels, while others may be lasting. Stress memory affects plants on molecular, biochemical, physiological, and morphological levels [168]. For instance, at the transcript level, “stress memory-related genes” can be characterized as those that display different expression patterns during repeated stress exposures compared to the initial stress event, whereas “non-memory genes” maintain consistent expression patterns regardless of whether the stress is experienced once or multiple times [176]. In *A. thaliana*, numerous cold-inducible genes exhibit a slight activation during reacclimation, with their expression levels remaining lower than those observed during the initial cold acclimation phase [177]. According to Byun et al. [177], the memory of cold stress can be retained due to an increased photosynthetic efficiency, altered lipid metabolism, increased calcium signalling, and the accumulation of protective proteins synthesized during initial cold acclimation, as well as changes in the signal transduction pathways influenced by these pre-existing protective proteins. Moreover, many changes are occurring on the molecular level, and they include different histone modifications [168]. Also, proteins can act as elements of plant stress memory. Some groups of proteins maintain increased levels after the cold acclimation and deacclimation of *A. thaliana*. Among them, there are stress-related proteins: LTI29, COR78, and TIL [178]. According to the authors, plants deacclimated quickly to complete growth and development, while simultaneously protecting themselves from possible temperature decreases.

Based on the studies of Stachurska et al. [122], it should be emphasized that cultivars with better stress memory should also be more tolerant to deacclimation, or at least should be able to reacclimate more quickly. This issue is important in the aspect of climate change and seems significant for the future, which will require introducing cultivars that are characterized by a better tolerance to deacclimation conditions and/or a better reacclimation capacity. According to [179], it would be even more beneficial for plants to maintain an adequate and effective reacclimation capacity, even after an extended period of deacclimation. Moreover, a critical factor for the winter survival of plants is deacclimation tolerance and/or the ability to reacclimate quickly [161]. Also, research related to reacclimation capacity is crucial, as global climate change intensifies, episodes with higher deacclimating temperatures are more frequent, and decreases in frost tolerance threaten crop plants.

## 4. Conclusions

Based on the literature review, it appears that deacclimation is not only just a reversal of changes induced by cold acclimation. Rather, it is a distinct process in which certain metabolic adjustments are partially retained. Such mechanisms may be associated with the plants’ reacclimation capacity, which allows them to respond faster in cases of re-exposure to low temperatures. In the aspect of intensifying climate change, it is becoming crucial to focus on cultivating plants that are more resilient to temperature fluctuations occurring during cycles of deacclimation and reacclimation. However, the detailed mechanisms underlying these processes still require further studies.

## Figures and Tables

**Figure 1 ijms-26-11080-f001:**
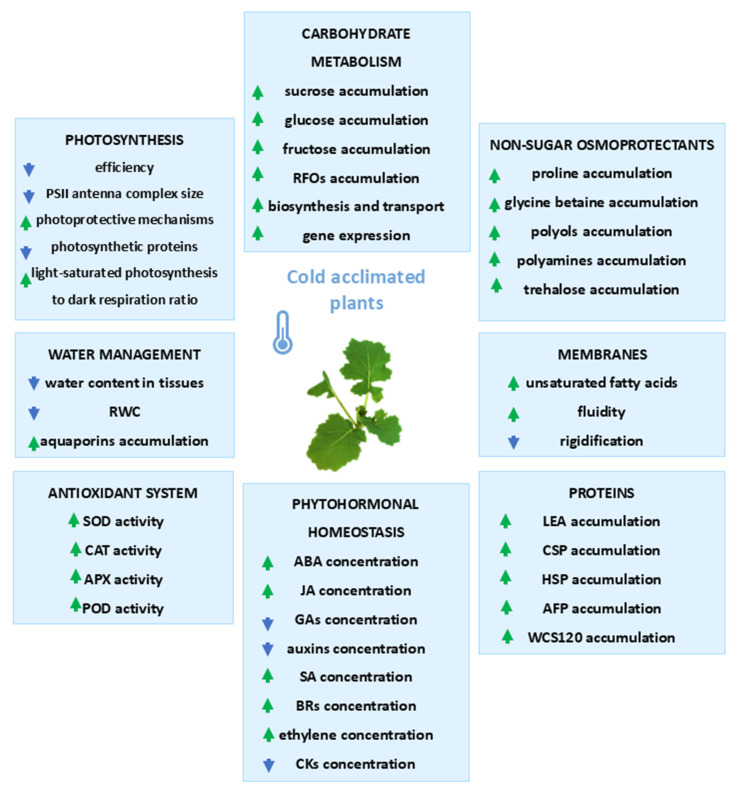
Effects of cold acclimation on plant’s metabolism. Green arrow—increase in the parameter; blue arrow—decrease in the parameter; based on the literature cited in Section 1.

**Figure 2 ijms-26-11080-f002:**
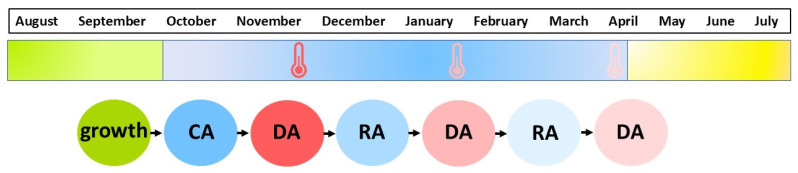
Schematic representation of temperature fluctuations during the winter season in Central Europe (Poland, Czech Republic, Slovakia), showing periods of temporary warming that can interrupt cold acclimation and induce deacclimation in plants.

**Figure 3 ijms-26-11080-f003:**
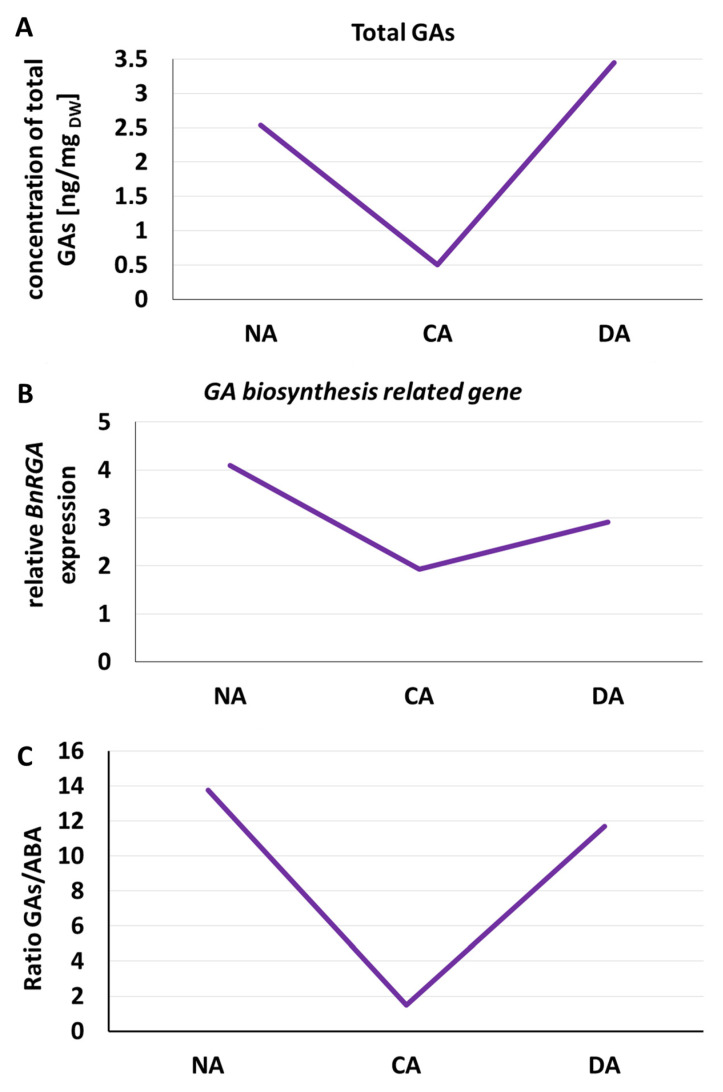
Visualization of the changes in the total content of active forms of gibberelins (GA1, GA3, GA4, GA5, GA6, GA7) (ng/mg_DW_) (**A**), relative expression of GA biosynthesis-related gene (*BnRGA*) (**B**), and the changes in the hormonal balance between the active forms of the growth promoting and stress hormones (ratio GA1 + GA3 + GA4 + GA5 + GA6 + GA7/ABA) (**C**) in leaves of oilseed rape cv. Kuga grown under control (NA), cold acclimation (CA), and deacclimation (DA) conditions—based on the original data, which are available in [148].

**Figure 4 ijms-26-11080-f004:**
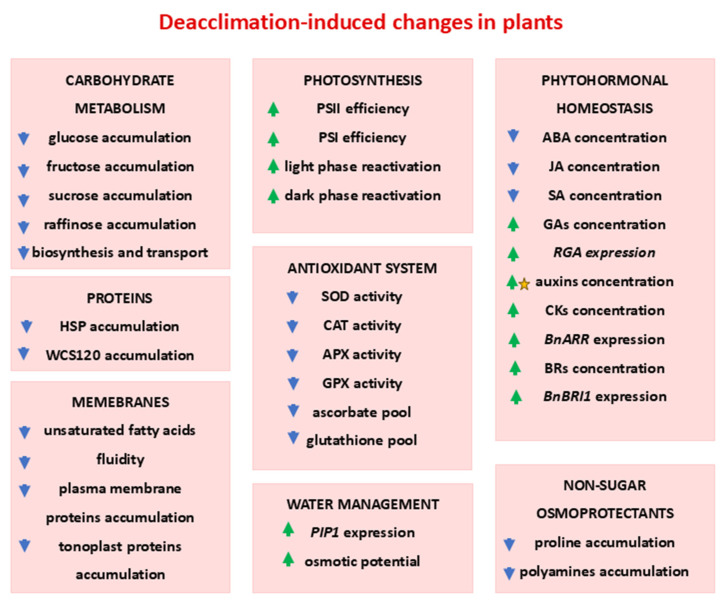
Effects of deacclimation on plant’s metabolism—comparison with cold acclimated plants. Green arrow—increase in the parameter; blue arrow—decrease in the parameter; asterisk—dependence on cultivar; based on the literature cited in Section 2.

**Figure 5 ijms-26-11080-f005:**
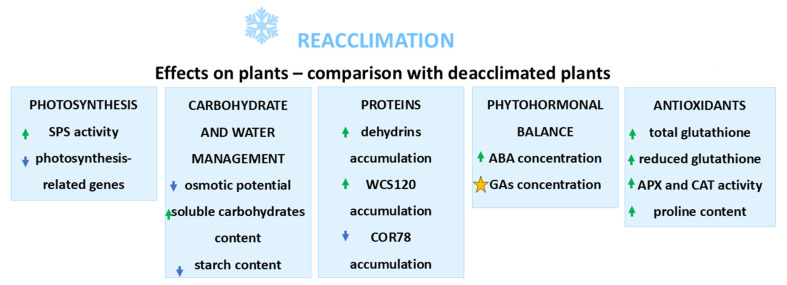
Effects of reacclimation on metabolism of plants—comparison with deacclimated plants. Green arrow—increase in the parameter; blue arrow—decrease in the parameter; asterisk—dependence on cultivar or temperature height; based on the literature cited in Section 3.

## Data Availability

No new data were created or analyzed in this study. Data sharing is not applicable to this article.

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
