# Peer review of "When Warm Breaks Cold: Understanding Deacclimations and Reacclimations Cycles as a Key to Winter Crop Resilience"

_ijms, 2025, doi:10.3390/ijms262211080_

Round 1
Reviewer 1 Report
Comments and Suggestions for Authors
Dear Authors,
Reviewer comments ijms-3980710
The review manuscript entitled „When cold breaks warm: Understanding deacclimations and reacclimations cycles as a key to winter crop resilience“ represents a valuable overview on the phenomena of cold acclimation, deacclimation and reacclimation in winter crops such as oilseed rape, wheat, and barley in temperate regions such as central Europe (Poland, Czechia, etc.) where warm spells induce deacclimation in cold-acclimated winter crops which provides a risk of efficient reacclimation under the following freezing events in late winter and early spring. The review manuscript focuses on the three key processes: cold acclimation, deacclimation, and reacclimation with a focus on key aspects including photosynthesis, carbohydrate metabolism, water management, plant membranes, antioxidants, proteins, and phytohormone homeostasis discussed with respect to each of cold acclimation, deacclimation, and reacclimation. The topic of the review manuscript partially overlaps with the recently published review by Kosová et al. (2025) Plant Physiol. Biochem. 220: 109541; doi: 10.1016/j.plaphy.2025.109541; however, in addition to winter Triticeae cereals, the authors also focus on winter oilseed rape as an important crop. The sectioning and the individual topics discussed are also different although close due to the relatedness of the topics discussed in both review manuscripts. However, I think that the present manuscript represents a vital complementary study to that by Kosová et al. (2025) thus I can recommend its publication in IJMS.
However, I have some important comments on the present manuscript which should be adequately addressed by the authors and which are provided below:
1/ The manuscript title: „When cold breaks warm:“ should be modified as follows: „When warm breaks cold“ since it is about warm spells during cold winter seasons which induce cold deacclimation in overwintering plants.
2/ Part 1.4. line 128: What is „water band index“?? I think that the term should be briefly explained in the text.
3/ In Figure 1 providing a scheme of the most common warm spells occurrence during winter season, the region for which this pattern is specific should be defined; e.g., central Europe (Poland, Czechia).
4/ I recommend the authors to combine Figure 2 and Figure 3 into a single Figure 2 with parts A, B, and C since all three graphs refer to the same literature source [146].
5/ In the headings, I recommend to modify „Sugar management“ to more scientifically rigorous „Carbohydrate metabolism“.
6/ Part 1.6. about cold acclimation-related proteins including dehydrins and HSP proteins such as HSP70 and HSP90 is mostly descriptive providing just literature overview. I think that it should be modified to provide critical discussion and formulate hypotheses on the potential roles of the proteins discussed in cold acclimation process.
7/ Line 479: The statement: „The aquaporin regulation during temperature stress is highly plastic and species-, genotype-, and isoform-specific.“ – this statement points to teh role of protein isoforms in plant responses to the processes of cold acclimation, deaclimation, and reacclimation. However, the topic of the role of proteoforms (protein isoforms and PTMs) in modulation of plant stress responses isnot discussed in the present study. I can recommend the authors to add this topic into their review and to add a reference on the review by Kosová et al. (2021) Front. Plant Sci. 12, 793113. https://doi.org/10.3389/fpls.2021.793113 on proteoforms into literature references.
8/ Lines 745-748: The text providing very interesting information about COR78 which is – unlike dehydrins – not reaccumulated under cold reacclimation lacks any relevant reference. References have to be added to this text demonstrating aletered dynamics of COR78 reaccumulation pattern under cold reacclimation!
9/ Formal comments on the text:
Line 299: Add a comma between the words ůand“ and „in the case of the expression of FeSOD and Cu/ZnSOD…“
Line 523: Replace the word „higher“ with either „enhanced“ or „increased“ in the statement:“….with deacclimated genotypes then more slowly retaining enhanced levels of WCS120 and exhibiting increased winter survival…“
Line 647: Replace the word „higher“ with either „enhanced“ or „increased“ in the statement: „Deacclimation is a revesible process, and deacclimated plants can regain enhanced frost tolerance in the process called reacclimation.“
Line 678: Replace the verb „related“ with either „affected“ or „determined“ in the statement: „Moreover, reacclimation is a process affecetd by cold stress memory and vernalization fulfillment.“ – reacclimation is not related to either cold stress memory or vernalization but the process of reacclimation is affected/ determined by cold stress memory and vernalization fulfillment.
Line 784: The reference on Figure 7 has to be corrected to Figure 4.!!
Final recommendation: Accept after a minor revision.
Author Response
We would like to thank the reviewer for taking the time to write this review. Their responses are attached in a PDF file.

Reviewer 2 Report
Comments and Suggestions for Authors
Julia et al. gathered the advances in deacclimation and reacclimation in this review. I should say it is a qualified work and it can help us to understand winter crop resilience fastly. I just have some minor suggestions that the authors must fix:
- I found that the references in this work seems not the newest, I hope you can add some paper that published in this year.
- I hope you can add the figures to show the main content of each subtitle.
- I hope you can add more molecular regulation advances.
- For the figures in text, I hope you can incease their qualities, them seems perfunctory and blurry.
I think the authors should adress my minor suggestions before it is published.
Author Response

(The authors gave the same response as above.)
